# Radiolabelled FGF-2 for Imaging Activated Fibroblasts in the Tumor Micro-Environment

**DOI:** 10.3390/biom14040491

**Published:** 2024-04-18

**Authors:** Valeria Bentivoglio, Filippo Galli, Michela Varani, Danilo Ranieri, Pallavi Nayak, Annunziata D’Elia, Andrea Soluri, Roberto Massari, Chiara Lauri, Alberto Signore

**Affiliations:** 1Nuclear Medicine Unit, Department of Medical-Surgical Sciences and of Translational Medicine, Faculty of Medicine and Psychology, “Sapienza” University of Rome, 00189 Rome, Italy; valeria.bentivoglio@uniroma1.it (V.B.); michela.varani@uniroma1.it (M.V.); pallavi.nayak@uniroma1.it (P.N.); chiara.lauri@uniroma1.it (C.L.); 2Department of Life Sciences, Health and Healthcare Professions, University “Link Campus University”, 00189 Rome, Italy; d.ranieri@unilink.it; 3Institute of Biochemistry and Cell Biology (IBBC), National Research Council of Italy (CNR), 00015 Monterotondo Scalo, Italy; annunziata.delia@cnr.it (A.D.); andrea.soluri@ibbc.cnr.it (A.S.); roberto.massari@cnr.it (R.M.); 4Unit of Molecular Neurosciences, University Campus Bio-Medico, 00128 Rome, Italy

**Keywords:** fibroblasts, tumor microenvironment, cancer imaging, nuclear medicine

## Abstract

Tumor associated fibroblasts (TAFs) play a key role in tumor growth and metastatization. TAFs overexpress different biomarkers that are usually expressed at low levels in physiological conditions. Among them are the fibroblast growth factor receptors (FGFRs) that bind the fibroblast growth factors (FGFs). In particular, the overexpression of FGFR-2c in tumors has been associated with advanced clinical stages and increased metastatization. Here, we developed a non-invasive tool to evaluate, in vivo, the expression of FGFR-2c in metastatic cancer. This is based on ^99m^Tc-labelled FGF-2. Methods: ^99m^Tc-FGF-2 was tested in vitro and in vivo in mice bearing allografts of sarcoma cells. Images of ^99m^Tc-FGF-2 were acquired using a new portable high-resolution ultra-sensitive gamma camera for small animal imaging. Results: FGF-2 was labeled with high specific activity but low labelling efficiency, thus requiring post-labeling purification by gel-filtration chromatography. In vitro binding to 2C human keratinocytes showed a Kd of 3.36 × 10^−9^ M. In mice bearing J774A.1 cell allografts, we observed high and rapid tumor uptake of ^99m^Tc-FGF-2 with a high Tumor/Blood ratio at 24 h post-injection (26.1 %ID/g and 12.9 %ID) with low kidney activity and moderate liver activity. Conclusions: we labeled FGF-2 with ^99m^Tc and showed nanomolar Kd in vitro with human keratinocytes expressing FGF-2 receptors. In mice, ^99m^Tc-FGF-2 rapidly and efficiently accumulated in tumors expressing FGF-2 receptors. This new radiopharmaceutical could be used in humans to image TAFs.

## 1. Introduction

In recent years, the role of the tumor microenvironment (TME) has emerged as a key-factor in the development and spread of cancer metastasis [1]. In the TME, tumor cells and stromal components interact with each other, leading to tumor progression and survival. One of the most abundant stromal components is tumor-associated fibroblasts (TAFs) [2]. During cancer pathogenesis, they are involved in the remodeling of the extracellular matrix (ECM) structure, allowing the interaction of the tumor cells with other cancer cells or with stromal cells through the secretion of chemokines, growth factors and cytokines [3,4]. TAFs have become an interesting target in oncological research due to the overexpression of different biomarkers [5]. Among them are fibroblast growth factor receptors (FGFRs), which bind fibroblast growth factors (FGFs).

The FGF family consists of 23 signaling peptides that are involved in biological processes, such as embryonic development, hematopoiesis and tissue regeneration [6]. They are expressed in almost all organisms, from nematodes to vertebrates, and are highly conserved in amino acid sequences of 120 amino acids (16–65% sequence identity) and gene structures [7].

In adults, FGFs act as homeostatic factors in cellular proliferation and tissue repair [8]. In particular, FGF-2 is an angiogenic factor that can also activate neovascularization during tumorigenesis. Moreover, it has oncogenic potential due to its mitogenic and differentiating effects [9].

FGF–FGFR signaling is mediated via direct binding of the growth factor to tyrosine auto-phosphorylation sites on the activated receptor, allowing the formation of a complex with other additional complements of signaling proteins. FGF signaling plays an important role during the epithelial–mesenchymal transition (EMT) during the development and progression of tumors [10]. During the EMT process, TGF-β induces the switch of the FGFR isoform of the epithelial cells, which typically do not bind FGF-2, to a sensitive isoform that binds FGF-2 and promotes cancer progression [11,12]. Indeed, the switch from the FGFR-2b epithelial isoform (which binds FGF-1, FGF-3, FGF-7 and FGF-10, but not FGF-2) to an FGFR-2c mesenchymal isoform (which binds FGF-2, FGF-1 and FGF-9) may drive EMT through enhanced FGF binding and signaling [13]. The switch from FGFR-2b to FGFR-2c has been observed in different carcinomas (oral squamous cell carcinoma, prostate, ovarian, bladder, breast, pancreatic, colorectal and non-small-cell lung cancer) [14]. It has been observed that the overexpression of FGFR-2c in tumors is associated with advanced clinical stages and increased metastasis. Furthermore, it is upregulated in patients who previously underwent anti-VEGF therapies, including the tyrosine kinase inhibitor (TKI) sorafenib, leading to the development of drug resistance [15,16]. Moreover, the plasmatic concentration of FGF-2 is increased in many cancers, such as leukemia, lung and breast cancer, often in a metastatic state. A recent study demonstrated that FGF-2 is involved in shifting tumor-associated macrophages (TAMs) toward the M2-phenotype (pro-tumorigenic) in the tumor microenvironment, as evidenced by comparisons between wild-type mice and mice genetically deficient in FGF [17]. These observations suggest the importance of developing a non-invasive tool to evaluate, in vivo, the expression of FGFR-2c in TAFs as well as TAMs, serving as a diagnostic/prognostic tool. To this aim, we conducted in vitro and animal experiments to determine the best protocol for radiolabeling human recombinant FGF-2 (hrFGF-2) with ^99m^Tc and to evaluate its potential to bind to FGFR-2-positive cell lines and image FGFR-2 tumors in vivo.

## 2. Materials and Methods

### 2.1. Conjugation

Recombinant human FGF-2 (17.2 kDa) was purchased from PeproTech (London, UK). For the reconstitution of lyophilized FGF-2, as advised by the data sheet of the product, the vial was centrifuged before opening and 1 mL of Tris-Buffer (Tris(hydroxymethyl)aminomethane) 5 mM (pH 7.5) was added to 1 mg of protein. Subsequently, through the use of a Zeba™ Spin Desalting Column 7K MWCO 2 mL (ThermoFisher, London UK), the protein samples were processed to remove salts and other small molecules (<1000 MW). After centrifugation at 2500× *g* rpm for 2 min, the purified sample was recovered and measured using a Bicinchoninic acid Assay (BCA, Thermo Scientific™ Pierce™, Waltham, MA USA). The protein concentration was determined against a standard reference of bovine serum albumin (BSA, Sigma-Aldrich, St. Louis, MO, USA). The labeling of FGF-2 was performed using an indirect method. Briefly, FGF-2 was conjugated with a heterobifunctional crosslinker, succinimidyl-6-hydrazinonicotinate hydrochloride (S-HYNIC, ABX, Radeberg, Germany) [18]. A total of 1 mg of HYNIC was dissolved in dimethylformamide (50 µL) (DMF; Sigma-Aldrich, St. Louis, MO, USA) and was incubated for 2 h in the dark at room temperature. To remove the excess linker from the modification reaction, the desalting procedure was conducted using a Zeba™ Spin column (ThermoFisher, London UK). Different HYNIC:protein molar ratios (1:10 and 1:20) were tested.

The amount of HYNIC bound to FGF-2 was assessed using a molar substitution ratio assay (MSR). This is a colorimetric assay utilizing a 0.5 mM solution of p-nitrobenzaldehdye (4-NB, dissolved in DMF) in 0.1 M 2-(N-morpholino) ethanesulphonic acid (MES) buffer, with a pH of 5.0. For the test, 10 µL of HYNIC-FGF-2 was added to 490 µL of solution, and the sample was incubated at 37 °C for 20 min before spectrophotometric analysis at 390 nm [19].

### 2.2. Radiolabeling with ^99m^Tc

Technetium-99m was obtained by elution using a ^99^Mo/^99m^Tc generator. HYNIC-FGF-2 (50 µL, 2 mg/mL) was labelled with different amounts of freshly eluted ^99m^TcO4 (NaCl 0.9%). The reaction was conducted in the presence of different amounts of co-ligand tricine (1, 10 or 20 mg in 50 µL) and with constant amount of stannous chloride (SnCl_2_) (10 µg in 1 µL) to evaluate the best labeling conditions [20]. Tricine (Sigma-Aldrich, St. Louis, MO, USA) was dissolved in distilled water and SnCl_2_ (Sigma-Aldrich, St. Louis, MO, USA) in purged HCl 0.1 M (mg/mL). The reaction solution was incubated for 10 min at room temperature and the labeling efficiency (LE%) and colloid percentages were evaluated using instant Thin Layer Chromatography (iTLC) and high-performance liquid chromatography (HPLC). To purify the ^99m^Tc-HYNIC-FGF-2 from the free ^99m^Tc, a PD-10 desalting column containing Sephadex G-25 resin (GE Healthcare, Uppsala, Sweden) was used.

### 2.3. Quality Controls

For iTLC, silica gel strips (Pall LifeSciences, Port Washington, NY, USA) were used as the stationary phase and NaCl 0.9% solution was used as the mobile phase for the determination of free pertechnetate (Rf = 0.9). For colloid determination (Rf = 0.1), we used a NH_3_:H_2_O:EtOH (1:5:3) solution as the mobile phase and silica gel strips coated with 5 mg/mL of bovine serum albumin (BSA) (Sigma-Aldrich, St. Louis, MO, USA). The iTLC strips were analyzed using a linear radio-scanner equipped with a collimated gamma-ray detector (Bioscan Inc., Poway, CA, USA) and each species was determined. The scan time was 3 min. The percentage of free pertechnetate was calculated as follows:%[free pertechnetate] = 100 × (Free Pertechnetate Area at front)/(Total activity)

The percentage of colloid was calculated through SG-BSA fiberglass sheets as follows:%[Colloid] = 100 × (Colloids Area at origin)/(Total activity)

With the obtained results, it was possible to calculate the LE%, as follows:LE (%) = 100 − [%Free pertechnetate + %Colloids]

HPLC was performed using a Gilson system, employing reverse-phase chromatography with a core-shell Kinetex C-18 column (5 mm diameter, 5 µm pore size, 250 mm length, Phenomenex, Torrance, CA, USA) with water for HPLC plus (Carlo Erba Reagents, Cornaredo (MI), Italy) with 0.1% TFA (Pump A) and Acetonitrile (ThermoFisher, London UK) with 0.1% of TFA (Pump B), using a gradient of 0–10 min at 0–95% B, 10–15 min at 95% B and 15–18 min at 95–5% B. The elution was conducted at a flow rate of 1mL/min, with 20 µL of the sample injected. The column effluent was monitored for UV absorbance using a dual wavelength spectrophotometric detector at 210 and 280 nm and using an on-line detector for radioactivity (Flow-Count, BIOSCAN, Eckert & Ziegler, Wilmington, MA, USA), connected in series. Aggregated ^99m^Tc-HYNIC-FGF-2 and ^99m^Tc-colloid was retained inside the column.

### 2.4. Stability to Challenge with Cysteine

The stability of the labeled peptide was tested using a cysteine challenge with different concentrations of cysteine (Sigma-Aldrich, St. Louis, MO, USA). The cysteine concentrations were 0.0001 mg/mL, 0.001 mg/mL, 0.01 mg/mL, 0.1 mg/mL and 1 mg/mL in PBS (pH 7). ^99m^Tc-labeled FGF-2 (10 µL each vial) was incubated for 1 h at room temperature with cysteine. After incubation, the percentages of free radioisotope and labeled protein were measured using iTLC using the mobile phase described above.

### 2.5. In Vitro Cell Binding Assay

The affinity of the radiopharmaceutical and the capacity of the ^99m^Tc-FGF-2 to bind the receptor after the radiolabeling process was assessed in vitro using a competitive binding assay. The measurements were obtained using a LigandTracer™ (Ridgeview Instruments AB, Uppsala, Sweden) [21]. Two different cell lines were used: a human keratinocyte (HEK001) cell line (CRL-2404, ATCC, Manassas, VA, USA), which stably overexpress FGFR-2c (2C cells), and EV cells overexpressing FGFR-2b.

To obtain stable expression of FGFR2c, the human keratinocyte cell line HaCaT was transduced with the retroviral expression vector pBp-FGFR2c-WT (2C) (Addgene, plasmid #45699), with the pBABE-Puro empty vector (EV) (Addgene, Cambridge, MA, USA, plasmid #51070) as a control [22].

The cells were cultured in Dulbecco’s modified eagle’s medium (DMEM), supplemented with 10% fetal bovine serum (FBS) plus antibiotics, seeded in the defined area of a tilted cell dish, and incubated in a humified incubator at 37 °C and 5% CO_2_ for 24 h.

To perform the binding assay, a cell dish containing 2C cells was gently washed with PBS, then incubated with BSA solution at 2% containing a known aliquot of radiopharmaceutical. After 3 h, when maximum uptake was reached, the radiolabeled solution was replaced with culture medium without radiopharmaceutical to evaluate the release of radioactivity from the cells. The level of radioactivity retained by the cells was measured in real-time. At the end of the measurement a binding/release curve was drawn to calculate the Kd value [23].

As controls, binding assays were conducted with free ^99m^Tc with 2C cells and with ^99m^Tc-FGF-2 with EV cells.

### 2.6. In Vivo and Ex Vivo Experiments in Mice

All animal experiments were carried out in compliance with the local ethics committee and in agreement with national rules and EU regulations (Study 204/2018-PR). To perform the targeting studies, syngeneic models (allografts) were generated in female BALB/c mice (8 weeks old, obtained from Envigo, Indianopolis, IN, USA) by implanting 10^6^ J774A.1 cells (a macrophage cell line expressing FGFR2c and capable of binding human FGF-2 [17]) in Matrigel^®^ (BD-Biosciences, Bergen, NJ, USA) in their right thigh, which induced murine reticulum cell sarcoma. The cells were purchased from American Type Culture Collection (ATCC^®^ TIB-67™, Milan, Italy) and grown in ATCC-formulated Dulbecco’s Modified Eagle’s Medium supplemented with 10% of FBS at 37 °C and in 5% CO_2_.

After about 20 days from the inoculation, the tumors became palpable and the targeting experiments were performed. In each mouse, we injected ^99m^Tc-FGF-2 (100 µCi in 100 µL). At different time points (3 h and 24 h), three mice per group were sacrificed and their major organs (small and large bowel, kidneys, spleen, stomach, liver, muscle, bone, lungs and heart), blood and tumor samples were collected in vials and were weighed. The radioactivity from each vial was counted in a single-well γ-counter (PerkinElmer, Waltham, MA, USA). The percentage of injected dose per organ (%ID) and the percentage of injected dose per gram (%ID/g) were calculated.

### 2.7. In Vivo Imaging in Mice

Scintigraphic images were acquired using a high-resolution gamma camera (HRGC) based on a proprietary design [24]. This device involves a square-hole tungsten collimator paired with a pixelated scintillation array, which is optically coupled to an arrangement of silicon photomultipliers (SiPMs). A readout electronics module, specially designed for low power consumption, processed the signals from the SiPMs and fed a 4-channel analogue-to-digital conversion (ADC) card.

The collimator, made of pure tungsten, consisted of 200 μm thick septa arranged to form 2.4 × 2.4 mm^2^ square channels with a collimation length of 24 mm.

The scintillation assembly comprised an array of 20 × 40 CsI(Tl) crystals embedded in a white epoxy resin, which acts as a mechanical support and as an optical reflector. Each crystal, with overall dimensions of 2.4 × 2.4 × 5.5 mm^3^, was isolated from its neighbors using a reflector thickness of 0.2 mm.

This design guaranteed exact alignment between the collimators’ holes and the pixels of the scintillation assembly. A SiPM tile of 2 × 4 modules S13361-3050NE-08 (Hamamatsu Photonics Europe, Herrsching Germany) measured the scintillation light.

The readout electronics module exploits a charge division circuit (CDC) that splits the charge along row or column directions, whereby two chains of resistors encode the outputs into four channels [25]. A dedicated four-channel 12-bit ADC system with a USB interface samples the output signals. This system ensures a maximum count rate of about 40 kCounts/s.

This high-resolution camera was used to acquire dynamic images (1 frame/min for 90 min) in 3 BALB/c mice bearing murine reticulum cell sarcoma in the right thigh in order to evaluate the in vivo kinetic binding of ^99m^Tc-FGF-2 to tumor cells. For these studies, the animals were anesthetized with a mixture of 1.5% isoflurane and their body temperatures were monitored.

In another set of mice, which were used for the biodistribution studies as mentioned in Section 2.6, planar gamma camera images were also acquired before sacrificing the mice at 3 h or 24 h post-injection of ^99m^Tc-FGF-2.

## 3. Results

### 3.1. Radiolabeling with ^99m^Tc and Quality Controls

With a HYNIC:FGF-2 molar ratio of 10:1 and 20 mg tricine, we obtained an LE of 28 ± 6.3% in three consecutive experiments. Using a molar ratio of 20:1, we obtained an LE of 60.1 ± 21.5%. Indeed, the HYNIC:FGF-2 ratio was more important that the amount of tricine used since no significant difference was observed when using 1, 10 or 20 mg tricine (Figure 1). Therefore, we decided to use a tricine concentration of 400 mg/mL, which gave the highest LE. Therefore, 20 mg (50 µL) of tricine was incubated with 100 µg of pre-conjugated HYNIC-FGF-2 (2 mg/mL, 50 µL), 500 µg SnCl_2_ (10 mg/mL, 50 µL) and 5 mCi ^99m^Tc (in 50–200 µL).

As shown in Figure 2, the iTLC results demonstrated a very low percentage of free technetium and colloid, with a final LE of 86.33%.

Radiopharmaceutical purity, after PD10 purification, was evaluated using reverse-phase HPLC with a C18 column. The analysis showed the minimal presence of free ^99m^Tc on the radiogram and no significant other impurities on the chromatogram with 280 nm UV (Figure 3).

### 3.2. Stability to Challenge with Cysteine

The results of the cysteine challenge are shown in Figure 4. The labeling of ^99m^Tc-FGF-2 was stable up to a very high concentration of cysteine (0.1 mg/mL). We saw the presence of free ^99m^Tc form FGF-2 only with the highest concentration of cysteine (1 mg/mL).

### 3.3. Binding Assay

Experiments using the LigandTracer™ on 2C cells (keratinocytes overexpressing FGFR-2c) showed faster radiopharmaceutical uptake than in EV cells (keratinocytes that express FGFR-2b), reaching a plateau within 60 min and demonstrating slow dissociation from the cells. The calculated Kd for ^99m^Tc-FGF-2 on 2C cells was 3.36 × 10^−9^ M and the calculated Kd for EV cells was 3.46 × 10^−5^ (Figure 5). Free ^99m^Tc uptake by 2C cells was negligible.

### 3.4. In Vivo and Ex Vivo Experiments in Mice

As shown in Figure 6 and Figure 7, single organ counting at 3 h post-injection (p.i.) indicated the rapid clearance of radiolabeled FGF-2 from the bloodstream and its accumulation mainly in the liver and spleen. To a lesser extent, ^99m^Tc-FGF-2 was taken up by the kidneys and lungs. All these organs showed a reduction of activity by 24 h.

In contrast, tumor uptake was higher at 24 h than at 3 h p.i., indicating persistent accumulation over time. At 3 h post-injection, the Tumor/Muscle ratio was 4.0 as %ID/g, and at 24 h post-injection, it was 151.6. The Tumor/Blood ratio at 3 h was 4.1 and at 24 h post-injection, it was 26.1 (Table 1).

### 3.5. In Vivo Imaging in Mice

In vivo dynamic imaging showed the rapid and continuous accumulation of radiolabeled FGF-2 in the tumor within 90 min from i.v. injection. Liver, lungs and kidney clearance was rapid after the initial accumulation, as shown by the ex vivo results from the biodistribution experiments.

The in vivo planar images of mice, acquired 3 h or 24 h post-injection of ^99m^Tc-FGF-2, confirmed the biodistribution of ^99m^Tc-FGF-2 with uptake and retention in the liver and spleen, and to a lesser extent in the kidneys (Figure 8).

## 4. Discussion

Several studies have demonstrated that the switch from epithelial isoforms of fibroblast growth factor receptors to mesenchymal isoforms is frequently involved in epithelial–mesenchymal transition (EMT) and cancer progression [26]. The epithelial cell usually expresses the FGFR-3b isoforms, but during the EMT process, TGF-β induced the isoform switching of fibroblast growth factor receptors (FGFR-3c), causing the cells to become sensitive to FGF-2, inducing EMT and promoting cancer progression [5].

For this reason, FGF-2 seems to be an excellent candidate for the diagnosis of early tumor stages. Since the aim of this study was to develop a diagnostic approach, to synthesize this radiopharmaceutical, ^99m^Tc was used, which is a γ-emitting radionuclide that allows only scintigraphic localization inside the body without exerting any therapeutic effect.

The labeling reaction was performed by exploiting the bond of technetium to functionalized FGF-2 by inserting a chelating group (s-HYNIC, a bifunctional agent), which, in turn, binds the radionuclide of interest. The ability to incorporate aromatic hydrazine linkers on biomolecules using s-HYNIC (succinimidyl 6-hydrazinonicotinate acetone hydrazone, SANH), an amino-reactive reagent that directly converts the amino groups on biomolecules and surfaces to HYNIC groups [18], made it possible, with tricine as a co-ligand and SnCl_2_ as a reducing agent, to perform the radiolabeling reaction and synthesize ^99m^Tc-FGF-2. The high labeling efficiency obtained in in vitro experiments, without the presence of colloids, highlighted the correct choice of tricine and SnCl_2_ concentration. The excellent Kd obtained from real-time kinetic binding assays on 2C human keratinocytes suggests that the labeling reaction did not affect protein–receptor interactions.

In vitro, there were some difficulties in growing the J774A.1 cells as a monolayer for the LigandTracer experiments. These cells rapidly grow in multiple layers and are not ideal for in vitro binding assays. In contrast, 2C and EV cells can be grown in a monolayer.

For the in vivo experiments, tumor growth with xenografts of 2C or EV cells in BALB/c mice was sub-optimal. In contrast, J774A.1 allografts grew well and rapidly in BALB/c mice. Since both J774A.1 and 2C cells express high levels of FGFR2c and bind human FGF-2, we accepted the compromise of having in vitro data on a cell line and in vivo data with a different cell line, with both expressing the same receptor for FGF-2.

Finally, the in vivo biodistribution studies at 3 h and 24 h showed a decreasing uptake pattern for all tissues and organs, but the tumor increased uptake over time. The unexpected uptake in the lung can be explained by the formation of aggregates that will then rapidly dissolve into monomers, or by the presence of macrophages that represent an immunological barrier in lungs, particularly for large molecules [27,28]. The low accumulation of the radioisotope observed in the stomach indicates low release of the isotope from the radiolabeled protein, since free ^99m^TcO4- normally accumulates in this organ [29].

Another specific feature of TAFs is the expression of fibroblast activation protein (FAP). This is a serine protease known for its role during embryogenesis and tissue modelling. In adults, its expression is very low, but tumors may exhibit infiltration of TAFs overexpressing FAP. Furthermore, the presence of FAP correlates with a worse prognosis [30]. For these reasons, in recent years, small molecules that inhibit this protein have gained interest in detecting and treating tumors. Indeed, FAP-specific enzyme inhibitors (FAPIs) have demonstrated high specificity and affinity and good pharmacokinetic profiles in both pre-clinical and clinical phases [31,32,33,34,35]. Indeed, in the last 10 years, several new FAPI-based radiopharmaceuticals have been developed, with promising results for diagnostic and therapeutic purposes [36,37,38,39,40,41,42].

Despite the advantages that radiolabeled-FAPI could bring as a diagnostic/therapeutic tool, there are several pitfalls and limitations. For example, the presence of false-positive results in non-oncological fibrotic pathologies [43,44,45,46,47]. As compared to radiolabeled FAPI, radiolabeled FGF-2 might provide additional prognostic information in some types of tumors. Comparative studies are necessary to evaluate which of these two radiopharmaceuticals is more useful for different solid tumors. Indeed, our study must be considered as a proof of concept, although it has several limitations. We used a very limited number of mice, due to increasing difficulties in using animal models, and we tested only one tumor cell line. Therefore, more studies are required, encompassing different cancer types and with longer study times, to better clarify the dosimetric aspects of radiolabeled FGF-2.

## 5. Conclusions

In this study, we described the development of a new radiopharmaceutical based on the conversion of the amino groups of FGF-2 to HYNIC groups for binding ^99m^Tc, with tricine serving as a co-ligand. The labeling reaction was performed at room temperature and showed good stability over time. However, the labeling efficiency varied in different experiments, necessitating a final purification step to separate ^99m^Tc-FGF-2 from unbound ^99m^Tc.

In vivo, ^99m^Tc-FGF-2 showed high initial accumulation in the liver, spleen and lungs, followed by tumor localization with high Tumor/Muscle and Tumor/Blood ratios, particularly at 24 h post-injection.

Despite promising results, these initial experiments need confirmation with more experiments and different cancer cells. Other chelating agents for ^99m^Tc could also be tested in order to improve the labeling efficiency, thus avoiding the need for final purification from free-^99m^Tc.

As a proof of concept, FGF-2 seems to be an attractive candidate for the molecular imaging of solid tumors.

## Figures and Tables

**Figure 1 biomolecules-14-00491-f001:**
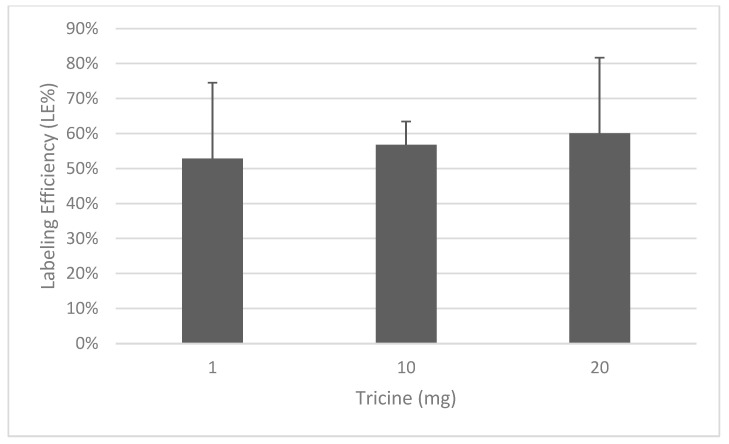
Labeling efficiency of FGF-2 with different concentrations of tricine. The HYNIC:protein ratio was always 20:1. These results show the means and standard deviations of three to four experiments.

**Figure 2 biomolecules-14-00491-f002:**
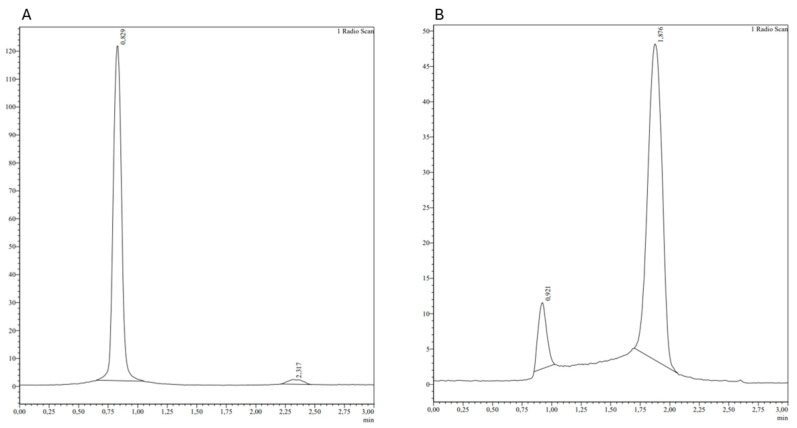
iTLC chromatograms of the highest LE formulation. (**A**) Chromatogram of the SG fiberglass sheet for the determination of free technetium; (**B**) chromatogram of the SG-BSA fiberglass sheet for the determination of colloid. The top of each peak represents the Retention time (Rt).

**Figure 3 biomolecules-14-00491-f003:**
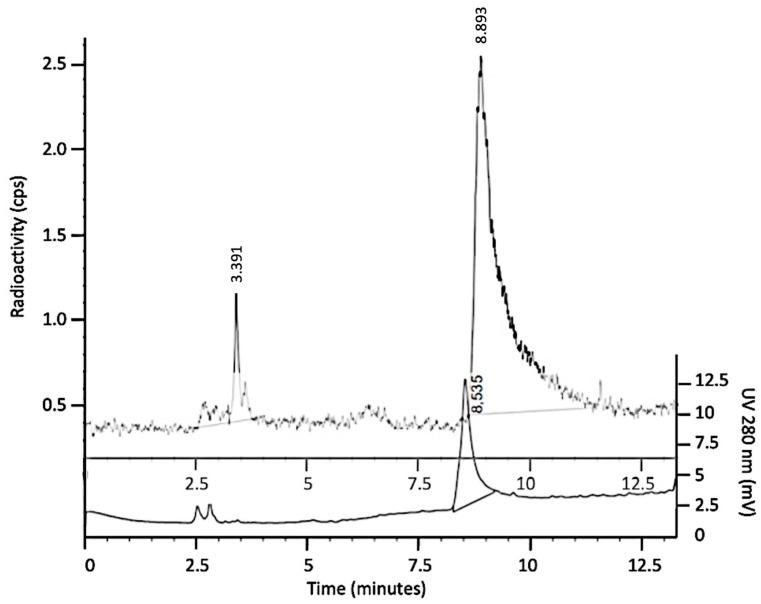
Radiogram (dotted line) and chromatogram (continuous line) of ^99m^Tc-FGF-2 using HPLC analysis after PD10 purification. Free ^99m^Tc eluted after 3.3 min (1.8%), whereas radiolabeled FGF-2 eluted after 8.5 min (98.2%).

**Figure 4 biomolecules-14-00491-f004:**
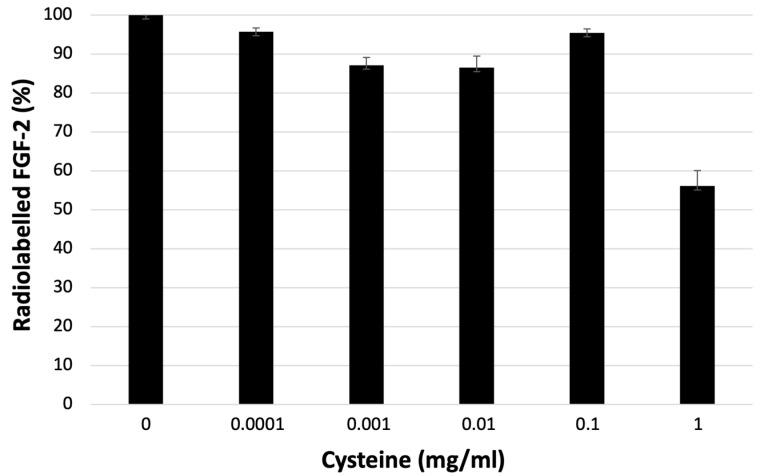
Cysteine challenge to test the stability of ^99m^Tc binding to the HYNIC-FGF-2 complex.

**Figure 5 biomolecules-14-00491-f005:**
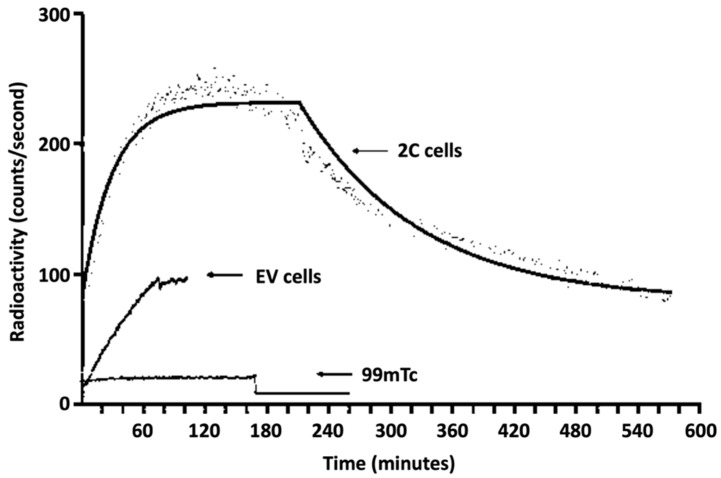
Real-time kinetic binding assay on human keratinocytes. The 2C cells indicate keratinocytes overexpressing FGFR-2c. The EV cells indicate keratinocytes that normally express FGFR-2b.

**Figure 6 biomolecules-14-00491-f006:**
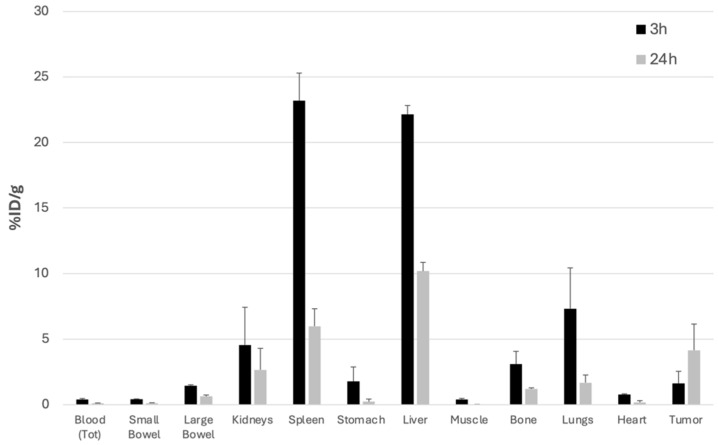
Biodistribution of ^99m^Tc-FGF-2 in BALB/c mice. The data represent ex vivo counts of each organ at different time points (3 h and 24 h). For each time point, three mice were sacrificed. The measured activity is expressed as the % of injected dose/g of tissue (%ID/g).

**Figure 7 biomolecules-14-00491-f007:**
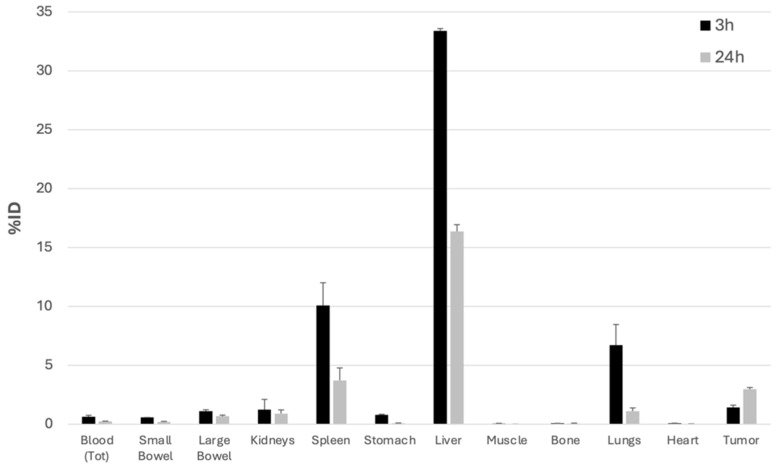
Biodistribution of ^99m^Tc-FGF-2 in BALB/c mice. The data represent ex vivo counts of each organ at different time points (3 h and 24 h). For each time point, three mice were sacrificed. The measured activity is expressed as the % of injected dose in the whole organ (%ID).

**Figure 8 biomolecules-14-00491-f008:**
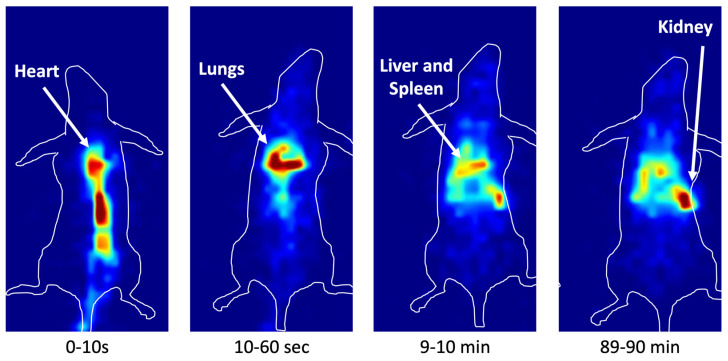
Dynamic images of ^99m^Tc-FGF-2 in normal BALB/c mice. The images were acquired with a frame rate of 1 image/minute for 90 min, except for the first frame of 10 s. This picture shows four representative frames at different time points, highlighting the rapid plasma disappearance of ^99m^Tc-FGF-2 and its metabolism by the liver and kidneys.

**Table 1 biomolecules-14-00491-t001:** Tumor/Muscle and Tumor/Blood ratios at different time points.

		3 h	24 h
Tumor/Muscle	%ID/g	4.0	151.6
	%ID	20.7	341.8
Tumor/Blood	%ID/g	4.1	26.1
	%ID	2.3	12.9

## Data Availability

The data presented in this study are available on request from the corresponding author.

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
