# Peer review of "Radiolabelled FGF-2 for Imaging Activated Fibroblasts in the Tumor Micro-Environment"

_biomolecules, 2024, doi:10.3390/biom14040491_

Round 1

Reviewer 1 Report

Comments and Suggestions for Authors

Imaging the tumour microenvironment can be an essential tool for the early diagnosis of many malignancies. Fibroblast activation protein is a type-II membrane bound glycoprotein specifically expressed by activated fibroblasts almost exclusively in pathological conditions including arthritis, fibrosis and cancer. FAP is overexpressed in cancer-associated fibroblasts located in tumour stroma. Recently the most promising candidate for nuclear diagnosis is a FAP-inhibitor, based on the structure developed by the researchers of the University of Antwerp. Any alternative molecular target on this field can improve the usefulness of the CAF-based diagnostic efforts. Therefore, the selection of the topic is highly important and innovative.

FGF-2 brings a big potency into the CAF visualization. The authors of this work operate with the common, well known techniques (for example the HYNIC-labelling, using the molar ratio of 20:1 is also similar, like the optimal ratio of proteins (for example SA) etc. But I guess, one innovative idea should be enough, especially, if it leads to a good result. But the application of a not really chemosensitive conjugation method will risk that the next bathes will be a bit different from the current one, and this circumstance deduct from the value of the establishment.

I have some other comments also, mainly in the form of the presentation:

line 124: "C-18 column (5 mm, 5 μ m, 250 × 4.6 mm, Phenomenex, Torrance, CA, USA" Phenomenex a big company with different C-18 columns. Please, specify. What kind of parameter is 5 mm?

In the section 2.3 iTLC method was described to determine colloids. In the Results 3.1 section there is no any data about the measurement. Please, supply. The determination of the RP with HPLC will show only components which can be remove from the column, therefore TLC verification has importance.

The publication of figure 5. and 6. are not complete, missing the number of the animals. Please, supply.

It is strange for me, that the abbreviation of "Fibroblast Growth Factor 2" was placed to the line 320. Somewhere else (f.e. line 325) this abbreviation is not uniform. I suggest a minor revision of the entire tex.

Independently from the remarks I consider this work suitable for the publication.

Author Response

Replies enclosed

Reviewer 2 Report

Comments and Suggestions for Authors

This manuscript describes the development of a novel technetium-99m labelled probe "99mTc-FGF-2" targeting tumor associated fibroblasts (TAF). This study was well-designed and performed. However, it seems to remain unclear to confirm if the 99mTc-FGF-2 can work as a SPECT imaging probe for evaluating TAFs because the in vivo data shown in this manuscript were inadequate for confirming the author's conclusion. Therefore, this manuscript should be accepted after revising in accordance with the comments as below.

Major comments

1. In the section "3.1 Radiolabelling with 99mTc and quality control", the author mentioned that the best HYNIC: protein ratio and tricine concentration were 20:1 and 400 mg/mL, respectively, but the reason remains unclear. Thus, the detailed data that made the authors concerned above should be added.

2. 2C cells were used as FGFR2c-highly expressing cells in the in vitro study, while J774A. 1 cells were used as FGFR2c-highly expressing cells in the in vivo study. Why was the cell line changed in the in vivo study?

3. Does the 99mTc-FGF-2 have a high affinity to mouse FGFR2c? To evaluate the biodistribution of 99mTc-FGF-2 in normal organs, the information must be necessory.

4. In the in vivo study, the mice xenografted FGFR2c low-expressing tumor cells should also be used to evaluate the non-specific accumulation of 99mTc-FGF-2 in the tumor tissues.

5 In the in vivo biodistribution study, the evaluation time points were set at longer times (3 h and 24 h), while those were set at shorter times (within 90 min) in the planer imaging study. To consider that the FGF-2 protein has a relatively longer biological half-life, the imaging study should also be performed at a longer time (at least 3 h).

Minor comments

6. The last sentence about the purification step in the  "2.3 Quality control" section should be moved to the "2.2 Rabelling with 99mTc" section.

7. In the "2.5 In vitro cell binding assay" section, detailed information on EV cells should be included like that on 2C cells.

8. In the "2.6 In vivo and ex vivo experiments in mice" section, a detailed number of the J774.A.1 cells xenografted in mice should be written.

9. In the supplementary files, it is hard to find the tumor tissues at a glance. Thus, the tumor and main organs, such as the liver and kidney, should be pointed  within the movie.

10. In the discussion section, the authors mentioned 99mTc-FGF-2 showed better biodistribution compared with FAPI-based probes,  but the reason why the author thought about that remains unclear. Thus, more detailed comparison and discussion should be added.

Author Response

Replies enclosed
